# Clinical features and novel technologies for prehospital detection of intracerebral haemorrhage: a scoping review protocol

Mohammed Almubayyidh ![ORCID] ,[1,2] Ibrahim Alghamdi,[1,3] Adrian Robert Parry-Jones,[1,4] David Jenkins ![ORCID] [5]

¹Division of Cardiovascular Sciences, The University of Manchester, Manchester, UK
²Department of Aviation and Marine, Prince Sultan Bin Abdulaziz College for Emergency Medical Services, King Saud University, Riyadh, Saudi Arabia
³Department of Emergency Medical Services, College of Applied Medical Sciences, Khamis Mushait Campus, King Khalid University, Abha, Saudi Arabia
⁴Manchester Centre for Clinical Neurosciences, Northern Care Alliance NHS Foundation Trust, Salford, UK
⁵Division of Informatics, Imaging and Data Science, The University of Manchester, Manchester, UK

**Correspondence to**
Mohammed Almubayyidh;
mohammed.almubayyidh@postgrad.manchester.ac.uk

## ABSTRACT

**Introduction** The detection of intracerebral haemorrhage (ICH) in the prehospital setting without conventional imaging technology might allow early treatment to reduce haematoma expansion and improve patient outcomes. Although ICH and ischaemic stroke share many clinical features, some may help in distinguishing ICH from other suspected stroke patients. In combination with clinical features, novel technologies may improve diagnosis further. This scoping review aims to first identify the early, distinguishing clinical features of ICH and then identify novel portable technologies that may enhance differentiation of ICH from other suspected strokes. Where appropriate and feasible, meta-analyses will be performed.

**Methods** The scoping review will follow the recommendations of the Joanna Briggs Institute Methodology for Scoping Reviews as well as the Preferred Reporting Items for Systematic Reviews and Meta-Analyses extension for Scoping Reviews checklist. A systematic search will be conducted using MEDLINE (Ovid), EMBASE (Ovid) and CENTRAL (Ovid). EndNote reference management software will be used to remove duplicate entries. Two independent reviewers will screen titles, abstracts and full-text reports according to prespecified eligibility criteria using the Rayyan Qatar Computing Research Institute software. One reviewer will screen all titles, abstracts and full-text reports of potentially eligible studies, while the other reviewer will independently screen at least 20% of all titles, abstracts and full-text reports. Conflicts will be resolved through discussion or by consulting a third reviewer. Results will be tabulated in accordance with the scoping review's objectives along with a narrative discussion.

**Ethics and dissemination** Ethical approval is not required for this review, as it will only include published literature. The results will be published in an open-access, peer-reviewed journal, presented at scientific conferences and form part of a PhD thesis. We expect the findings to contribute to future research into the early detection of ICH in suspected stroke patients.

## STRENGTHS AND LIMITATIONS OF THIS STUDY

⇒ No time of publication restrictions will be applied to comprehensively map the area of interest.
⇒ The review will be limited to English language publications only.
⇒ A formal risk of bias assessment will not be performed.

## INTRODUCTION

### Early detection of intracerebral haemorrhage

Intracerebral haemorrhage (ICH) has the highest mortality rate of any stroke subtype.[1] The 1-year case fatality of ICH is estimated to be around 50%, with nearly half of those deaths occurring within 72 hours of onset, predominantly due to neurological complications.[2 3] The risk is greatest in this early stage, as the haematoma can expand during the first few hours after the onset of symptoms.[4] Consequently, ongoing clinical trials of different treatment approaches are targeting the very early stages of the disease process, with the aim to improve outcomes for patients with ICH by reducing haematoma expansion.[5–7] A delay in interventions for ICH may lead to poor patient outcomes.[2] As treatment options appear highly time-dependent,[8] the early recognition of symptoms and the rapid delivery of interventions are crucial for effective management.

The majority of strokes occur at home, meaning that prehospital personnel are typically the first to make clinical contact with suspected stroke patients.[9] These personnel play a crucial role in the early triage and management of these patients. The initial diagnosis of stroke is based on clinical presentation and stroke recognition instruments, followed by neuroimaging confirmation in the hospital. Brain imaging using CT and MRI remains the gold standard for detecting stroke subtypes and accurate recognition of stroke subtypes prior to imaging is not felt to be possible.[10] The current American Heart Association and American Stroke Association guidelines state 'No existing clinical decision scale can differentiate ICH from other diseases with high sensitivity or specificity in the absence

of neuroimaging', but a systematic review of the literature was not presented and novel technologies were not discussed.[11]

The accurate recognition of stroke may be challenging for prehospital personnel due to the heterogeneity of clinical presentations, time constraints and the lack of accurate diagnostic technology in this setting.[12] Stroke treatment has become increasingly complex with critically time-sensitive procedures, such as intravenous thrombolysis, mechanical thrombectomy or neurosurgical operations only available in specialised hospitals (eg, hyperacute stroke unit).[13] It is thus desirable to not only differentiate stroke from non-stroke but to identify subtypes (ICH vs ischaemic stroke) and eligibility for interventions, ensuring patients are transported directly to a centre with the capability to deliver the appropriate interventions. Prehospital recognition of patients with ICH will facilitate the very early delivery of treatments to prevent haematoma expansion, such as intensive blood pressure reduction, anticoagulant reversal and haemostatic agents, which may potentially improve outcomes.[7 14] Nevertheless, it is important to note that such interventions must be thoroughly studied in adequately powered and well-designed clinical trials to determine their efficacy and safety. The ability to diagnose ICH in the ambulance easily and inexpensively would facilitate such clinical trials.

### Clinical features and emerging technologies to detect and classify ICH

Many clinical features of ICH are also features of other stroke types though some have been described that may help differentiate ICH from other suspected stroke patients.[15] For example, patients with spontaneous ICH tend to be younger than those with ischaemic stroke.[15] Also, the presence of hypertension and headache is more common on admission in patients with ICH; meanwhile, a history of transient neurological deficits, hyperlipidaemia and atrial fibrillation is more common in patients with ischaemic stroke.[16] These features were used to develop clinical prediction rules to distinguish ICH from ischaemic stroke.[16 17] Nevertheless, the diagnostic accuracy of these rules are low and may not be applicable to patients presenting within a few hours of the onset of symptoms or during the prehospital phase.[15 18]

Recent literature has proposed new approaches that could assist prehospital personnel recognise stroke subtypes, opening new ways to diagnose stroke and provide treatment earlier.[12] These include, but are not limited to, the development of new stroke screening tools, blood biomarkers technologies and new imaging modalities.[12 13 19–26] Many of these advances are still in the early stages of development; therefore, further investigation is required to determine whether they can detect stroke sufficiently and differentiate between stroke types, particularly within the first few hours of symptoms onset and their potential use in prehospital care.

### Objectives

The purpose of this scoping review is to summarise and describe the current state of knowledge regarding the early identification of ICH in suspected stroke patients. First, we aim to identify early clinical features that can help distinguish ICH patients from other stroke types and non-stroke diagnoses. Second, we aim to explore portable technologies that can be used by prehospital personnel to detect ICH. Finally, we will determine whether meta-analyses are appropriate and feasible based on the homogeneity of findings.

### METHODS
### Protocol design

This study will follow the guidelines of the Joanna Briggs Institute (JBI) Methodology for Scoping Reviews.[27 28] The Preferred Reporting Items for Systematic Reviews and Meta-Analyses extension for Scoping Reviews (PRISMA-ScR) checklist will be followed to ensure that all suggested items are reported (online supplemental appendix 1).[29] This protocol will be reported using the PRISMA-ScR checklist and JBI guidelines.

### Inclusion/exclusion criteria

The population, concept, context (PCC) framework is used to establish eligibility criteria, as suggested by the JBI.[27 28] Table 1 summarises the PCC framework of this protocol.

### Population

Included studies will describe adult patients (aged ≥16 years) with suspected stroke or intracranial haemorrhage confirmed by either CT or MRI scans. We will exclude studies of children (aged <16 years) from this review due to their different presentations, diagnostic challenges, risk factors and recurrence risk.[30] Additionally, we will exclude studies that combined the clinical features of spontaneous ICH patients with other aetiologies, such as subarachnoid haemorrhage, without describing them separately.

### Concept

We will first examine the concept of distinguishing spontaneous ICH from other stroke types and non-stroke diagnoses using early clinical features collected at the initial prehospital assessment or within the first 24 hours of symptoms onset at hospital admission. Second, we will explore the possibility of detecting ICH using portable devices. Studies should report the clinical characteristics of their included patients, including but not limited to, demographic data, signs and symptoms at onset, vital signs (eg, blood pressure, heart rate and blood glucose) and medical histories. Studies that evaluate novel portable devices will be required to report the accuracy of diagnosis (eg, sensitivity, specificity and/or discrimination) after testing the device on ICH patients. Where appropriate, we will evaluate the applicability of using

**Table 1** Inclusion and exclusion criteria

| PCC | Included | Excluded |
|---|---|---|
| Population | ► Studies of adult patients (aged ≥16 years) with suspected stroke or intracranial haemorrhage. | ► Studies of children (aged <16 years). ► Studies combining the clinical features of ICH with those of other aetiologies. |
| Concept | ► Studies reported early clinical features of suspected stroke patients, collected at prehospital assessment or within 24 hours of symptoms onset at hospital admission. ► Studies tested portable technologies to detect ICH. | ► Studies reported clinical features of patients after 24 hours of symptoms onset; or if they investigated clinical features associated with ICH without comparing them against other suspected stroke cases. ► Studies that did not report on the diagnostic performance of the tested technology; or examined advances in conventional detection modalities; or that tested the technology only on phantoms or animal models. |
| Context | ► Prehospital and in-hospital studies. | ► Primary care studies. |

ICH, intracerebral haemorrhage; PCC, population, concept, context.

the evidence identified in prehospital care. Studies that report the clinical features of patients after 24 hours of symptoms onset or only examined ICH patients' clinical features without comparing them to other stroke or non-stroke diagnoses will be excluded from this review. Also, we will exclude studies that examine advances in conventional detection modalities, such as CT or MRI, or tested the technology only on phantom or animal models.

### Context

The context of this scoping review is the prehospital and in-hospital care setting. It will include sources that report on healthcare professionals, including physicians, paramedics, nurses and others who provide patient care in a variety of settings, such as emergency departments, intensive care units and ambulance facilities. Primary care studies will not be included in this review. Further, there will be no restrictions on the country of study, ethnicity, gender or socioeconomic status.

### Sources of information and search strategy

In this scoping review, we will include data from primary research studies. The search will be limited to human studies, and for resource reasons, will be restricted to studies published in English. Conference abstracts, commentaries, surveys, case reports, animal studies and articles in languages other than English will be excluded. The search strategy was developed based on specific keywords that were combined using the Boolean operators ("AND", "OR"), which can be found in online supplemental appendix 2. This search strategy will be applied to MEDLINE via Ovid, EMBASE via Ovid and CENTRAL via Ovid, from the date of inception to August 2022. Finally, we will review the reference lists of the included articles to identify other additional studies, thus minimising the likelihood of missing any eligible publication.

### Study records and selection process

Following the database search, all identified studies will be imported into EndNote and Rayyan Qatar Computing Research Institute (QCRI) software to remove duplicates, as well as to facilitate screening and collaboration. The selection of studies for inclusion in the review will be conducted by two independent reviewers in two stages. Initially, the titles and abstracts will be screened, and then a full-text review will be conducted. The first reviewer (MA) will screen the titles and abstracts to remove studies that do not meet the predefined inclusion/exclusion criteria before retrieving the full-text of eligible studies. The second reviewer (IA) will independently screen a random selection of at least 20% of the titles and abstracts. Any discrepancies about the suitability of any of the papers will be resolved through discussion, or by consulting a third reviewer (ARP-J or DJ) as required. The first reviewer (MA) will then review the full-text reports against the inclusion and exclusion criteria. The second reviewer (IA) will independently screen at least 20% of the full-text reports as well, and any discrepancies will be resolved through discussion, or by consulting a third reviewer (ARP-J or DJ) as required. We will record the reasons for exclusion after the fulltext assessment of works that do not meet the inclusion criteria. Detailed information about the search and the study inclusion process will be presented in the final scoping review report with a PRISMA-ScR flow chart.[29]

### Data extraction process and items

The general characteristics of the included studies will be summarised using basic publication data (eg, authors, publication year, country, research design, study environment), study population information (eg, sample size, stroke types, clinical features collected), details of the detection device (eg, technology used, diagnostic accuracy, time to results, potential limitations) and key findings of the studies. The summary of results will be presented in different tables in accordance with the scoping review's objectives. The tabulated results will be accompanied by a narrative description of the evidence. Furthermore, the results will be described in terms of how they address the objectives of this review.

## Data synthesis and quality assessment of studies

The analysis of ICH detection methods relies on the data obtained from each study. Given that this is a scoping review, a narrative synthesis will be used to summarise the findings of the included studies. Furthermore, since there is no expectation for a bias assessment on the PRISMA-ScR checklist,[29] we will not conduct a formal assessment of the risk of bias of the included studies.

## Patient and public involvement

In this scoping review, the data collected are based on previously published studies, so neither the patients nor the general public will be involved.

## Ethics and dissemination

The results of this review will be published in an open-access, peer-reviewed journal, and the findings will be presented at scientific conferences and will be part of a PhD thesis. We expect that the study findings will be useful in future to develop research into the early detection of ICH in suspected stroke patients by prehospital personnel, and to help them make field assessments and decisions about the treatment and destination triage of these patients. This scoping review will only investigate published literature; therefore, ethical approval is not required.

## CONCLUSION

As treatment options appear highly time-dependent, the effective management of ICH patients requires the early recognition of clinical features, delivery of appropriate interventions and correct transport decisions to a hospital with neurosurgical care for eligible patients. Although suspected stroke patients may have similar or overlapping clinical features, some early findings may suggest the presence of ICH. To facilitate the diagnosis, several new technologies and approaches have been proposed to enhance the prehospital care of suspected stroke. Our study will inform future research by mapping the literature to help prehospital personnel distinguish ICH from other suspected stroke patients to improve the field assessment, treatment and destination triage to the appropriate level of care.

**Contributors** MA is the primary author of this document under the supervision of ARP-J and DJ. DJ conceptualised the idea for the review. MA designed the search strategy and drafted the protocol paper, incorporating the suggestions put forward by ARP-J, DJ and IA. All authors critically reviewed the protocol and approved the final manuscript before final submission for peer review.

**Funding** This study is funded by King Saud University, Saudi Arabia, through the Saudi Arabian Cultural Bureau in the United Kingdom.

**Competing interests** None declared.

**Patient and public involvement** Patients and/or the public were not involved in the design, or conduct, or reporting, or dissemination plans of this research.

**Patient consent for publication** Not applicable.

**Provenance and peer review** Not commissioned; externally peer reviewed.

peer-reviewed. Any opinions or recommendations discussed are solely those of the author(s) and are not endorsed by BMJ. BMJ disclaims all liability and responsibility arising from any reliance placed on the content. Where the content includes any translated material, BMJ does not warrant the accuracy and reliability of the translations (including but not limited to local regulations, clinical guidelines, terminology, drug names and drug dosages), and is not responsible for any error and/or omissions arising from translation and adaptation or otherwise.

**ORCID iDs**
Mohammed Almubayyidh http://orcid.org/0000-0003-3266-1236
David Jenkins http://orcid.org/0000-0001-6687-3507

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
