## [Reviewer comments · BMJ Open]

ARTICLE DETAILS

TITLE (PROVISIONAL)	Clinical features and novel technologies for pre-hospital detection of intracerebral haemorrhage: a scoping review protocol
AUTHORS	Almubayyidh, Mohammed; Alghamdi, Ibrahim; Parry-Jones, Adrian; Jenkins, David

VERSION 1 – REVIEW

REVIEWER	Adam Oostema Michigan State University, Emergency Medicine
REVIEW RETURNED	05-Jan-2023

GENERAL COMMENTS	Thank you for the opportunity to review this manuscript. The authors present a protocol for a scoping review. The goal of the review is to survey the published literature to describe methods for differentiation of intracerebral hemorrhage from other types of stroke without the use of conventional CT imaging (such as clinical features or other point of care tests). The manuscript is generally well-written, and the topic is of interest to regional stroke systems attempting to optimize hospital destination decisions made by EMS when transporting patients with suspected stroke. Nevertheless, this topic has been examined in the past. Clinical characteristics alone have consistently performed poorly in differentiating hemorrhagic from ischemic stroke (Runchey JAMA 2010) and clinical scores such as the Siriraj score have been only modestly successful (Bhardwaj J Stroke Cerebrovasc Dis 2022). I am not aware of any new large clinical studies examining this issue to add to this literature. The more interesting aspect of this review might involve looking at new technologies for stroke detection in the prehospital setting, however another review directly addressed this question last year (Chennareddy et al BMC Emergency Medicine 2022). Introduction In the current world, hospital destinations should likely be selected based on the severity of stroke symptoms rather than underlying diagnosis. LVO patients and ICH tend to present with severe symptoms and both conditions are likely best served in comprehensive stroke centers. There is not currently any direct evidence that more accurate prehospital diagnosis of hemorrhagic stroke would allow for “widespread and cost-effective early treatment.” I am not aware of a significant body of research that demonstrates improved ICH outcomes when anticoagulation or BP management are initiated earlier; in fact, INTERACT2 did not show a differential effect for patients randomized within 4 hours of symptom onset vs later. Considering this, and since this review is focused on
--

	whether ICH patients can be distinguished from other kinds of stroke, I would suggest softening or eliminating some of the speculation about how more accurate prehospital diagnosis would impact outcomes. On the other hand, the introduction could focus more on the findings of previous literature on this topic to explain why another review is needed. Is there newly published data? Did prior evaluations of this literature have important limitations addressed by this review? Methods The inclusion and exclusion criteria to be applied to identify relevant studies could be stated more definitively. I find the PCC organization nice for explaining rationale, but somewhat difficult to follow in determining exactly what will be included/excluded from this analysis. Perhaps a table of inclusion and exclusion criteria would be helpful? Study designs – will there be any criteria based upon study design? Target population – There is also some language in the “Concept” section that suggests that studies of ICH patients that do not also contain a comparison group of other stroke or non-strokes will be excluded. This seems to contradict the statement that studies of confirmed ICH patients will be included. Exposures – I think it would be useful to include a complete list of clinical/demographic exposure variables that will be targeted for abstraction from these studies, perhaps as an online supplement. Will the review also include previously developed clinical tools for stroke recognition? Similarly, could specific categories of diagnostic tests or devices that may be included in this review be delineated? Conclusion Treatment options for ICH are not time-dependent. Individuals are not excluded from certain forms of treatment based upon timing of arrival. As above, the efficacy of treatment may be time-dependent to some degree, but this has not yet been clearly demonstrated. I would suggest emphasizing that this survey of the literature will explore the feasibility of clinical or point-of-care options for recognition of ICH and refrain from speculating upon how that might impact outcomes.
--	--

REVIEWER	Craig Anderson George Institute for Global Health, The George Institute for Global Health
REVIEW RETURNED	07-Feb-2023

GENERAL COMMENTS	This manuscript reports a scoping review to plan for more formal meta-analysis to (i) identify demographic and clinical features that distinguish acute intracerebral haemorrhage (ICH) from acute ischaemic stroke in the pre-hospital (ambulance) setting, and (ii) determine the predictive value of novel imaging and blood biomarker tools to do the same in this context. Although there have been many hospital-based studies undertaken for this purpose, there is accumulated data from trials and observational studies that will allow a fresh perspective of this approach in the pre-hospital setting, and particular as there is a focus on early initiation of medical treatments to improve outcome from ICH. The investigators will register their protocol and use standard review and quantification techniques, with efforts to reduce bias in the
--

	inclusion. There are some minor issues that need addressing:  1) Although treatments for ICH are likely to be time-dependent, this is not proven so perhaps better to say 'might' rather than 'would' allow... in the introduction of the Abstract; and 'appear' rather than 'are' in line 6 of para 1 of the Introduction 2) it is uncertain why the authors have restricted themselves to only English language publications 3) I am not sure why there is no formal evaluation of publication bias even though a meta-analysis is not being undertaken 4) I wonder if the authors have considered subtyping the type of ICH, and even the morphology of the haematoma where there is good data to indicate those more likely to have ongoing bleeding than others
--	--

VERSION 1 – AUTHOR RESPONSE

Reviewer #1:

C1: Thank you for the opportunity to review this manuscript. The authors present a protocol for a scoping review. The goal of the review is to survey the published literature to describe methods for differentiation of intracerebral hemorrhage from other types of stroke without the use of conventional CT imaging (such as clinical features or other point of care tests). The manuscript is generally well-written, and the topic is of interest to regional stroke systems attempting to optimize hospital destination decisions made by EMS when transporting patients with suspected stroke.

Nevertheless, this topic has been examined in the past. Clinical characteristics alone have consistently performed poorly in differentiating hemorrhagic from ischemic stroke (Runchey JAMA 2010) and clinical scores such as the Siriraj score have been only modestly successful (Bhardwaj J Stroke Cerebrovasc Dis 2022). I am not aware of any new large clinical studies examining this issue to add to this literature. The more interesting aspect of this review might involve looking at new technologies for stroke detection in the prehospital setting, however another review directly addressed this question last year (Chennareddy et al BMC Emergency Medicine 2022).

R1: Thank you very much for this comment. The clinical features of ICH were previously examined and compared only to those with ischaemic stroke using hospital-based studies (Runchey et al., JAMA 2010), and the authors concluded that their findings may not be applicable to patients presenting to an emergency setting within a few hours of the onset of symptoms. Hence, our study will examine the concept of distinguishing ICH from other stroke types and non-stroke conditions (e.g., stroke mimics) using early clinical features collected at the initial pre-hospital assessment or within the first 24 hours of symptoms onset at hospital admission. We have now included this information in the revised manuscript.

On the other hand, the study by Chennareddy et al., 2022 (BMC Emergency Medicine), did a limited literature search, as they did not include EMBASE or CENTRAL, and therefore some relevant studies may have been missed. Furthermore, the authors did not elaborate extensively on how the devices performed in detecting or differentiating ICH specifically, and some of the evaluated studies included children in their analyses. Given that, our study will focus in greater depth on the ability of technologies to detect ICH in adults.

C2: In the current world, hospital destinations should likely be selected based on the severity of stroke symptoms rather than underlying diagnosis. LVO patients and ICH tend to present with severe symptoms and both conditions are likely best served in comprehensive stroke centers. There is not currently any direct evidence that more accurate prehospital diagnosis of hemorrhagic stroke would allow for “widespread and cost-effective early treatment.” I am not aware of a significant body of research that demonstrates improved ICH outcomes when anticoagulation or BP management are

initiated earlier; in fact, INTERACT2 did not show a differential effect for patients randomized within 4 hours of symptom onset vs later. Considering this, and since this review is focused on whether ICH patients can be distinguished from other kinds of stroke, I would suggest softening or eliminating some of the speculation about how more accurate prehospital diagnosis would impact outcomes.

R2: Thank you for raising the key issue of the potential clinical utility of pre-hospital diagnosis of ICH. We agree that strong evidence does not currently exist supporting very early intervention in ICH. However, the key, hyperacute therapeutic target in ICH is haematoma expansion, and it is well established that the risk of expansion is higher earlier in the disease course (Al-Shahi Salman et al., *Lancet Neurol* 2018). For this reason, ongoing trials of haemostatic agents are targeting very early in the disease process. For example, the FASTEST trial (NCT03496883) is aiming to deliver activated factor VIIa within 120 min of symptom onset. Acute BP lowering reduces haematoma expansion (Moullaali et al., *J Neurol Neurosurg Psychiatry* 2022), and there is evidence to support greater benefit for very early treatment in hospital (Li et al., *Ann Neurol* 2020). Whilst use of GTN patches in the ambulance appears not to be of benefit to a broader population of suspected stroke patients (RIGHT-2 Investigators, *Lancet* 2019), the RIGHT-2 trial authors speculate that this may be related to a harmful effect of nitrates specifically in ICH. Further pre-hospital testing of other antihypertensives in ICH would be facilitated by pre-hospital diagnosis. Finally, reversal of anticoagulation reduces haematoma expansion (Steiner et al., *Lancet Neurol* 2016) and evidence supports greater benefit with earlier delivery (Kuramatsu et al., *JAMA* 2015). Surgery targets a different aspect of the pathophysiology of ICH and earlier treatment again may be of greater benefit (Sondag et al., *Ann Neurol* 2020). We agree that transfer to comprehensive stroke centres with neurosurgery may be of benefit for ICH, but this remains to be proven. We have reformulated some of the statements to make it clearer that our research will focus on the identification of ICH in the pre-hospital setting, and such an approach may have the potential to improve patient outcomes.

C3: The introduction could focus more on the findings of previous literature on this topic to explain why another review is needed. Is there newly published data? Did prior evaluations of this literature have important limitations addressed by this review?

R3: We are grateful for your constructive suggestions. In addition to the above response to C1, we have clarified this in the revised manuscript by adding a sentence to the introduction: "These features were used to develop clinical prediction rules to distinguish ICH from ischaemic stroke. Nevertheless, the diagnostic accuracy of these rules are low and may not be applicable to patients presenting within a few hours of the onset of symptoms or during the pre-hospital phase". We also added more recent references in the introduction section that may further support the need for this review (page 5, line 21).

C4: The inclusion and exclusion criteria to be applied to identify relevant studies could be stated more definitively. I find the PCC organization nice for explaining rationale, but somewhat difficult to follow in determining exactly what will be included/excluded from this analysis. Perhaps a table of inclusion and exclusion criteria would be helpful?

R4: We thank the reviewer for this important suggestion. We have added a table to the revised manuscript (Table 1), summarising the inclusion and exclusion criteria of this analysis.

C5: Will there be any criteria based upon study design?

R5: Thank you for your comment. We will include all articles that meet the inclusion criteria, regardless of the study design chosen by researchers. This will enable us to determine which research designs are commonly used and/or can be applied in the future.

C6: There is also some language in the "Concept" section that suggests that studies of ICH patients that do not also contain a comparison group of other stroke or non-strokes will be excluded. This seems to contradict the statement that studies of confirmed ICH patients will be included.

R6: We thank the reviewer for this observation. In the revised manuscript, we have clarified that studies should evaluate the clinical features of ICH in comparison with other suspected stroke patients, whereas studies that tested portable technologies to detect ICH are not restricted by this comparison.

C7: I think it would be useful to include a complete list of clinical/demographic exposure variables that will be targeted for abstraction from these studies, perhaps as an online supplement.

R7: We thank the reviewer for this comment. The aim of this review is to identify clinical features and therefore we have not specified a set of variables to be targeted; rather, we aim to investigate any potential variable described in the literature. Predefining variables for abstraction could bias results. Nevertheless, we have briefly described the potential variables that will be targeted for abstraction from each study under the "Concept" and "Data extraction process and items" sections.

C8: Will the review also include previously developed clinical tools for stroke recognition? Similarly, could specific categories of diagnostic tests or devices that may be included in this review be delineated?

R8: Thank you for this comment. We are not specifically looking for these tools, but the search could potentially capture some of them. Our study aims to report on any evidence found on clinical predictors of ICH in studies of suspected stroke patients. Additionally, we have now referenced some of the potential eligible studies that tested portable technologies/devices to detect ICH under the section entitled "Clinical features and emerging technologies to detect and classify ICH".

C9: Treatment options for ICH are not time-dependent. Individuals are not excluded from certain forms of treatment based upon timing of arrival. As above, the efficacy of treatment may be time-dependent to some degree, but this has not yet been clearly demonstrated. I would suggest emphasizing that this survey of the literature will explore the feasibility of clinical or point-of-care options for recognition of ICH and refrain from speculating upon how that might impact outcomes.

R9: Thank you very much for your comment. In addition to the above response to C2, the current ESO guidelines state "In patients with hyperacute (<6 hours) intracerebral haemorrhage, we suggest lowering blood pressure to below 140 mmHg (and to keep it above 110 mmHg) to reduce haematoma expansion", excluding ICH patients from intensive BP lowering who present beyond 6 hours of symptom onset (Sandset et al., Eur Stroke J 2021). Further, it is possible that earlier treatment in the pre-hospital setting may improve outcomes, although it must first be tested in clinical trials that are adequately powered and well-designed. However, the ability to diagnose ICH in the pre-hospital setting easily and cheaply would pave the way for such trials. As mentioned in R2, we have added necessary statements in the revised manuscript to make this point clearer.

Reviewer #2:

C1: Although treatments for ICH are likely to be time-dependent, this is not proven so perhaps better to say 'might' rather than 'would' allow... in the introduction of the Abstract; and 'appear' rather than 'are' in line 6 of para 1 of the Introduction

R1: Thank you very much for your comment. We have amended the revised manuscript accordingly.

C2: It is uncertain why the authors have restricted themselves to only English language publications

R2: Thanks for your insightful comment. For resource reasons, we will restrict our study to English-language publications, and we have now included this information in the revised manuscript.

C3: I am not sure why there is no formal evaluation of publication bias even though a meta-analysis is not being undertaken

R3: We thank the reviewer for this excellent point. The risk of bias will not be assessed since the purpose of this scoping review is to provide an overview of the existing literature, and not to critically appraise the included articles. This is also consistent with the PRISMA guidelines for reporting on scoping reviews (Tricco et al., 2018; Ann Intern Med). Hence, we have labelled our review as scoping and not systematic. However, we will include a discussion of any limitations found in the included studies in the final scoping review report.

C4: I wonder if the authors have considered subtyping the type of ICH, and even the morphology of the haematoma where there is good data to indicate those more likely to have ongoing bleeding than others

R4: This is a very interesting comment. In fact, we do not expect that eligible studies will classify patients based on the location and morphology of ICH but if this is the case, we will report these findings. As a result, this has been added to our extraction form.

VERSION 2 – REVIEW

REVIEWER	Craig Anderson George Institute for Global Health, The George Institute for Global Health
REVIEW RETURNED	22-Mar-2023
GENERAL COMMENTS	The authors have adequately addressed reviewer concerns